# Compensation Mechanisms May Not Always Account for Enhanced Multisensory Illusion in Older Adults: Evidence from Sound-Induced Flash Illusion

**DOI:** 10.3390/brainsci12101418

**Published:** 2022-10-21

**Authors:** Heng Zhou, Xiaole Liu, Junming Yu, Chunlin Yue, Aijun Wang, Ming Zhang

**Affiliations:** 1Department of Psychology, Research Center for Psychology and Behavioral Sciences, Soochow University, Suzhou 215123, China; 2School of Physical Education and Sport Science, Soochow University, Suzhou 215021, China

**Keywords:** sound-induced flash illusion (SiFI), resting-state functional magnetic resonance imaging (rs-fMRI), regional homogeneity (ReHo), fission and fusion illusions

## Abstract

Sound-induced flash illusion (SiFI) is typical auditory dominance phenomenon in multisensory illusion. Although a number of studies have explored the SiFI in terms of age-related effects, the reasons for the enhanced SiFI in older adults are still controversial. In the present study, older and younger adults with equal visual discrimination were selected to explore age differences in SiFI effects, and to explore the neural indicators by resting-state functional magnetic resonance imaging (rs-fMRI) signals. A correlation analysis was calculated to examine the relationship between regional homogeneity (ReHo) and the SiFI. The results showed that both younger and older adults experienced significant fission and fusion illusions, and fission illusions of older adults were greater than that of younger adults. In addition, our results showed ReHo values of the left middle frontal gyrus (MFG), the right inferior frontal gyrus (IFG) and right superior frontal gyrus (SFG) were significantly positively correlated with the SiFI in older adults. More importantly, the comparison between older and younger adults showed that ReHo values of the right superior temporal gyrus (STG) decreased in older adults, and this was independent of the SiFI. The results indicated that when there was no difference in unisensory ability, the enhancement of multisensory illusion in older adults may not always be explained by compensation mechanisms.

## 1. Introduction

Many studies have demonstrated that sensory processing is mainly carried out in the form of multisensory integration [1,2,3], which is vital for understanding sensory processing. Multisensory integration is defined as a basic sensory process in which information from different senses is integrated into a single sensation [4]. The reliability of the signals associated with a given feature helps us when we integrate multisensory information to perceive our surroundings in our daily lives [4]. The information reliability hypothesis proposes that the dominant modality provides the most reliable information for multisensory integration [5]. Sound-induced flash illusion (SiFI) is a multisensory illusion caused by multisensory integration in which auditory information is dominant and preferentially processed over other sensory information [4,6,7].

Two forms of the SiFI exist, namely, fission and fusion illusions. When participants are presented with a visual stimulus and two auditory beeps, one visual stimulus is mistakenly reported as two, which is a condition known as the fission illusion [7,8]. Accordingly, when two visual flashes and one auditory stimulus, two visual stimuli are perceived as one, and such a condition is known as the fusion illusion [9,10]. Many studies have explored factors that influence the SiFI to further understand the underlying mechanism. Specifically, the characteristics of visual [9] and auditory stimuli [9], cognitive factors [9,11], experience [12,13] and individual differences [14,15] affect the generation of the SiFI. Among these factors, the individual variability related to the participants’ susceptibility to the SiFI cannot be ignored. For example, although studies based on large samples demonstrated that the average incidence of the SiFI was 50% [15], the probability of perceiving the illusion varied [4], ranging from 3% to 86% among individuals [16]. Studies have also revealed that older adults are more likely than younger adults to produce the SiFI, compared with younger adults, the fission illusion and the fusion illusion are both greater among older adults [17,18,19]. Numerous previous studies have shown that superior temporal gyrus (STG) is recognized as a key brain region for audio-visual tasks and multisensory integration [20], and the stronger the activation of the STG, the stronger the susceptibility to the SiFI [21,22,23]. Moreover, such an increase in age-related susceptibility to the SiFI has been supported by large population-based studies [24,25], this suggests that older adults may have greater multisensory illusion [26] and stronger neural activity in the STG.

The reason that older adults are more susceptible at multisensory illusion remains unknown. Many studies suggested that the enhanced multisensory illusion of the older adults may be a sensory compensation mechanism [27,28,29]. According to the Inverse Effectiveness [30], the reduced effectiveness of a single sensory signal can promote multisensory integration, this has been shown in younger adults to induce integration when low intensity audiovisual stimuli are used, but not when high intensity audiovisual stimuli are used [31]. For older adults, when single sensory functions decline, the human brain can enhance its ability to integrate information from multisensory modalities to compensate for the lack of single sensory information processing [27]. ERP study has also found that the amplitude of P1 induced by audiovisual stimuli was smaller than the sum of single visual and auditory stimuli, and the amplitude of P1 induced by audiovisual stimulation was smaller in older adults. The latency of N1 induced by audiovisual stimuli was significantly earlier than the sum of single visual and auditory stimuli, and the latency of N1 was earlier in older adults [29], this also supported the compensation mechanism for visual and auditory processing in older adults.

However, not all studies supported the idea that increased multisensory illusion was due to old adults’ compensation for unisensory ability. Previous studies have shown that older adults have enhanced prestimulus β-band activity in SiFI tasks, which may be related to perceptual priors [32]. Hirst et al., (2007) found that when older and younger adults had no difference in behavioral responses to a unisensory stimulus, the older adults’ audiovisual integration was still greater than younger adults, indicating that the compensation mechanism mentioned above could not fully explain the phenomenon [25]. Not only that, Using The Irish Longitudinal Study of Ageing data, Hirst et al., (2020) found that the visual gain of older adults with eye diseases did not differ from that of controls, and that the angular gyrus, rather than the primary sensory cortex, which is associated with higher cognitive functions, played an important role in age-related SiFI [33,34]. Therefore, we thought enhanced multisensory illusion did not always result from compensation for unisensory ability. And the purpose of this study was to find evidence to support this idea at the behavioral and neurobiological levels.

Resting-state functional magnetic resonance imaging (rs-fMRI) was often used to investigate the intrinsic resting neural activity of psychological mechanisms. Among rs-fMRI, regional homogeneity (ReHo) values assumed that the voxels in a functional brain region were more time-homogeneous in a characteristic state, and the voxel-level ReHo values were obtained by calculating the time series of a particular voxel and its neighboring voxels using Kendall consistency coefficient calculation [35,36]. ReHo values have been widely used to investigate individual variance in multisensory working memory and other cognitive tasks [37,38]. Previous studies have suggested that ReHo values could reflect individual differences in behaviors and cognition [38,39]. Therefore, we used ReHo values to reflect the differences between younger and older at the neurobiological level.

According to the perspective of compensation mechanism, the enhancement of multisensory illusion in older adults was to compensate for the decreased sensitivity of unisensory ability. Therefore, when there was no discrepancy between the unisensory ability of younger and older adults, if older adults can still show the enhancement of the multisensory illusion, it may not be explained by the compensation mechanism. Using the classical SiFI paradigm, the current study explored whether multisensory illusion was enhanced in older adults when there was no discrepancy between the unisensory ability of younger and older adults. Therefore, participants were asked to perform a unisensory flash discrimination task before the formal experiment to determine whether the flashes occurred once or twice, only higher performing participants were able to proceed to the formal experiment. Additionally, ReHo values were used to explore whether the differences in neural activity between the two groups were related to differences in the SiFI. We hypothesized that even though older and younger adults had similar basic ability to detect flashes, they still behaved differently in the illusion conditions, with greater SiFI illusions in older than in younger adults, moreover, the differences in neural activity between the two groups was independent of the SiFI.

## 2. Materials & Methods

### 2.1. Participants

In order to ensure that the participants had good ability to distinguish the flashes, we did a pre-experiment, which one or two visual flash stimuli were randomly presented. Participants were told to perform a task to judge visual flash stimuli appeared once or twice in each trial, and each condition included 80 trials. Based on previous research, participants with a hit rate of more than 80% were allowed to take the formal experiment [11,16,40]. Previous study has shown that female predict the higher susceptibility to the SiFI than male [24], so we recruited only female participants to make it easier to observe the SiFI. Finally, 27 younger females (age: *M* = 20.67, *SD* = 2.15) and 30 older females (age: *M* = 61.37, *SD* = 3.88) were recruited. G*Power 3.1.9.2 was used to estimate the statistical power by performing a sensitivity test of the between factors identified by the *F* test [41,42]. The analysis parameters were as follows: 0.05 was ɑ err prob, 0.80 was power (1- β err prob), 57 was total sample size, numbers of groups and measurements were 2. The final output was η^2^ = 0.27. This sample size was appropriate to test sufficient effect sizes.

All the participants had normal or corrected vision, and all participants self-reported that they had never taken part in a similar experiment, and signed an informed consent form according to the criteria of the Helsinki Declaration. The Ethics Committee of the Department of Psychology at Soochow University approved the conduct of this study. The resting-state scan was performed before the task. During the scan, participants remained relaxed, closed their eyes and did not do any intentional thinking. After a brief scan, they performed the SiFI task outside the scanner.

### 2.2. Apparatus and Procedure

All the experimental stimuli were presented in a View Sonic P220f VS10284 model which had a 1024 × 768 pixels screen resolution and a 60 Hz refresh rate. The visual flashes were programmed to appear on a black background by display Presentation software (version 20.0, Neurobehavioral Systems Inc., Albany, CA, USA). The visual flash was a white disk with a radius of 2°, rendered at a visual angle 5° below the “+” in the center of the screen for 17 ms. This is because when auditory beep stimulation is accompanied by visual flash stimulation, the illusion effect is greatest when visual flash stimulation is located in the peripheral vision [43]. The auditory beep stimulus was presented with a was 75 dB, a rendering time of 7 ms using a head-mounted iron triangle headset (ATH-WS99) and a frequency of 3.5 kHz. Participants were asked to tap on a mouse to make their judgments while sitting on a stationary chair 59 cm from the screen.

For the sake of description, trial types were indicated by abbreviations. F for flash, B for beep, experimental conditions can be expressed as F1, F1B1, F1B2, F2, F2B1 and F2B2. “F1B1” represents the trial in which an auditory beep stimulus and a visual flash stimulus occur simultaneously, and “F1B2” represents the trial in which a visual flash stimulus and two auditory beep stimuli occur. In F1B1, F1B2, F2B1, and F2B2, the first auditory beep stimulus and the first visual flash stimulus occurred at the same time. The intervals between the two visual flashes and the two auditory beeps were 66 ms and 76 ms, respectively. There were 80 trials under each experimental condition, i.e., 480 trials in total. The stimuli under 6 experimental conditions were randomly presented in the experiment, with intertrial intervals (ITI) ranging from 400 ms to 700 ms at a step size of 100 ms. The entire experiment consisted of 4 blocks, each blocks was presented pseudorandom, each of which included 120 trials, after which the participants could rest. Participants were required to judge the number of visual flashes, ignore auditory beeps and complete the task by clicking the mouse. To ensure a balanced response, half of the participants were randomly selected to report one flash by clicking the left mouse button and two flashes by clicking the right mouse button, while the other half did the opposite (a representation of the stimulus is shown in Figure 1).

### 2.3. rs-fMRI

#### 2.3.1. Image Acquisition and Analysis

The MRI images were obtained by using a Philips 3.0T MRI scanner, and gradient echo plane imaging (EPI) was used to obtain resting state images of all participants. The scanning parameters were provided: 2000 ms was TR, 30 ms was TE, 220 mm × 220 mm was field of vision, 90° was turning angle, 64 × 64 was matrix, 36 was number of layers, 4 mm was thickness, 400 s was scanning time. Magnetization-prepared gradient echo (MPRAGE) was used to obtain 3D-T1 images. TR/TE: 8.5/3.4 ms, field of view: 240 mm × 240 mm × 150 mm, scanning matrix size: 256 × 256 × 256, flip angle: 12° [44].

#### 2.3.2. Data Preprocessing

Data Processing & Analysis for Resting-State Brain Imaging (DPABI), a technique developed by Yan et al. (http://rfmri.org/dpabi (accessed on 12 August 2022), was applied to analyze and process images on the MATLAB2013b platform [45]. After all participants’ DICOM images were converted to NIFTI images, the preprocessing of data was performed. First, delete the first 10 time points, and slice timing (the reference layer refers to the layer that situated at the intermediate time point during whole brain scan), then realign (participants who had frequency movement exceeding 2 mm or rotation exceeding 2° were excluded), and removed covariates (consisting of six head parameters, white matter signals and CSF signals). Then DARTEL space was utilized for standardization, and the functional image parameters which were attained by segmentation were standardized, which used the Montreal standard brain as the template. The voxel size was 3 mm × 3 mm × 3 mm. Then, the linear trend was removed, and the image was time-band pass filtered (0.01~0.08 Hz) to diminish low-frequency drift and high-frequency noise [46,47].

#### 2.3.3. ReHo Analysis

ReHo analysis were conducted by using the software DPABI, And the Kendall consistency coefficient of the time series of adjacent voxel groups of brain regions was calculated to obtain ReHo value. The greater ReHo values of a specific voxel are, the higher the local consistency of the rs-fMRI signal between neighboring voxels. A cubic cluster of 27 voxels was generally utilized, and centrosomes would obtain ReHo value from the assignment of each cube cluster. Then the ReHo image was smoothed to a 4 × 4 × 4 mm^3^ half-width method.

### 2.4. Statistical Analysis

#### 2.4.1. Behavioral Data Analysis

The mean hit rate was the ratio of the number of correct response times to the total number of response times under one condition. In the fission illusion and fusion illusion conditions, to explore whether there was a difference between younger adults and older adults, for the fission illusion, we conducted a 2 (age group: younger adults vs. older adults) × 2 (condition: F1B1 vs. F1B2) repeated-measures ANOVA, and for the fusion illusion, we conducted a 2 (age group: younger adults vs. older adults) × 2 (condition: F2B1 vs. F2B2) repeated-measures ANOVA. Post hoc *t* tests based on Bonferroni correction were also performed. We used η_p_^2^ for ANOVA and Cohen’s *d* for the *t* test to report the effect size. In addition, in order to supplement the results of the hit rate, we analyzed the behavior of the participants by signal detection theory. According to the calculation method mentioned by Keil (2020) [4], when no beep appears, two flashes are judged as hits in the F2 condition, and two flashes are judged as false alarms in the F1 condition. When the beep appears only once, one flash is judged as false alarms in the F2B1 condition, and one flash is judged as hits in the F1B1 condition. When the beep appears twice, two flashes are judged as hits in F2B2 condition, and two flashes are judged as false alarms in F1B2 condition. From these values, the sensitivity and response criterion can be calculated by the formula *d’* = z(Hit rate) − z(FA), $\ln\left(\beta \right)=\frac{z{\left(\mathrm{FA}\right)}^{2}-z{\left(\mathrm{Hit}\;\mathrm{rate}\right)}^{2}}{2}$, where z stands for the standard score.

#### 2.4.2. Correlation Analysis of ReHo Values and Likelihood of the SiFI

##### Older Adults Group

A voxel-wise correlation analysis was conducted to explore the correlation between ReHo values and the likelihood of the SiFI in older adults. The likelihood of the illusion was defined by the hit rate, likelihood of fission illusion = 1 − F1B2(hit rate), likelihood of fusion illusion = 1 − F2B1(hit rate). DPABI software was used [45,47]. To control the type I error in Monte Carlo simulations, the following parameters were used: The initial threshold on voxel level was set as *p* < 0.05, the cluster level was *p* < 0.05, 1000 two-tailed simulations, 4-mm FWHM filtration, 5-mm cluster connection radius (edge connected). AlphaSim correction was performed, and *p* < 0.05 was considered to indicate statistical significance.

##### Between-Group Comparisons

Using DPABI software [45], we performed two-sample *t* test on ReHo values of older and younger adults. To control the type I error in Monte Carlo simulations, AlphaSim correction was performed, the following parameters were used: The initial threshold on voxel level was set as *p* < 0.05, the cluster level was *p* < 0.05, 1000 two-tailed simulations, 4-mm FWHM filtration, 5-mm cluster connection radius (edge connected). We extracted the brain regions with differences between the two groups as the template mask, according to this template, the ReHo values of older participants were extracted, and Pearson correlation analysis was conducted between the ReHo values and the likelihood of illusion of older adults, *p* less than 0.05 indicated statistically significant.

## 3. Results

### 3.1. Behavioral Performance

*Hit rate* In the condition of visual flashes, younger adults had a hit rate of 0.95 (*SD* = 0.05) in the F1 condition and 0.94 (*SD* = 0.05) in the F2 condition, and older adults had a hit rate of 0.93 (*SD* = 0.05) in the F1 condition and 0.92 (*SD* = 0.05) in the F2 condition. To make sure there were no discrepancy between younger and older adults in their ability to detect flashes, we conducted a 2 (age group: older vs. younger adults) × 2 (condition: F1 vs. F2) repeated measures ANOVA. The results showed that the main effect of the condition was not significant, *F*(1, 55) = 2.46, *p* = 0.12, η_P_^2^ = 0.04. The main effect of the age group was not significant, *F*(1, 55) = 3.13, *p* = 0.08, η_P_^2^ = 0.05. The interaction between age group and condition was not significant, *F* < 1. The results suggested that both the younger and older adults had the ability to correctly distinguish the number of flashes, and there was no difference in basic visual discrimination between the two groups.

For the fission illusion, we performed a 2 (age group: younger adults vs. older adults) × 2 (condition: F1B1 vs. F1B2) repeated measures ANOVA. The results showed that the main effect of the condition was significant, *F*(1, 55) = 159.87, *p* < 0.001, η_P_^2^ = 0.74, and the hit rate of F1B1 (0.96) was significantly higher than that of F1B2 (0.51). The main effect of the age group was significant, *F*(1, 55) = 4.93, *p* = 0.031, η_P_^2^ = 0.08, and the hit rate of younger adults (0.78) was significantly higher than that of older adults (0.69). The interaction between age group and condition was significant, *F*(1, 55) = 4.50, *p* = 0.038, η_P_^2^ = 0.08. To further examine the results of the interaction, we conducted a simple effect analysis with Bonferroni correction. The independent sample *t* test showed that, in the F1B1 condition, there was no significance in the hit rate between the two age groups, *t* < 1. The hit rate of older adults (0.43) was significantly lower than that of younger adults (0.59) in the F1B2 condition, *t*(55) = 3.07, *p* = 0.016, Cohen’s *d* = 0.58. The above results indicated that no matter younger and older adults suffered from the fission illusion in the F1B2 condition, but older adults had significantly greater likelihood of the fission illusion than younger adults.

For the fusion illusion, we performed a 2 (age group: younger adults vs. older adults) × 2 (condition: F2B1 vs. F2B2) repeated measures ANOVA. The results showed that the main effect of the condition was significant, *F*(1, 55) = 91.61, *p* < 0.001, η_P_^2^ = 0.63, and the hit rate of F2B2 (0.96) was significantly higher than that of F2B1 (0.61). The main effect of the age group was not significant, *F*(1, 55) = 2.23, *p* = 0.141, η_P_^2^ = 0.04. The interaction between age group and condition was not significant, *F*(1, 55) = 1.56, *p* = 0.217, η_P_^2^ = 0.03. The above results indicated that in the F2B1 condition, both younger and older adults demonstrated the fusion illusion, but two groups had no significant difference in the likelihood of the fusion illusion (see Figure 2).

#### Signal Detection Theory Analysis

*Sensitivity (d’)* To supplement the mean hit rate results, we compared the *d’* of older and younger adults and found that when only flash stimuli were presented, there was no difference between the *d’* of younger and older adults, *t* (55) = 1.91, *p* = 0.062. In the F1B2 condition, the *d’* of older adults was significantly lower than younger adults, *t* (55) = 2.96, *p* = 0.005, Cohen’s *d* = 0.79. In the F2B1 condition, there was no difference between the *d’* of younger and older adults, *t* (55) = 1.66, *p* = 0.10. The above results indicated that there was no difference in basic visual discrimination between younger and older adults. Compared with younger adults, older adults were more sensitive to the fission illusion, but there was no difference in the fusion illusion between the two groups (see Figure 3).

*Criterion ln(β)* We compared Criterion ln(β) of younger and older adults. And we found that none of the conditions were significant. When only flash stimuli were presented, *t* < 1, in the F1B2 condition, *t* (55) = 1.23, *p* = 0.22, or in the F2B1 condition, *t* < 1. These results suggested that the Criterion ln(β) of younger and older adults were similar.

### 3.2. Older Adults Group

#### Correlation of ReHo Value and Likelihood of Illusion

For the fission illusion, ReHo values were analyzed by Alphasim correction (*p* < 0.05, cluster size > 130 voxels), the likelihood of the fission illusion was positively correlated with ReHo values in the left middle frontal gyrus (MFG). And for the fusion illusion (cluster size > 129 voxels), the likelihood of the fusion illusion had positively correlations with ReHo values in the right inferior frontal gyrus (IFG) and right superior frontal gyrus (SFG), as shown in Table 1 and Figure 4.

### 3.3. Between-Group Comparisons

#### 3.3.1. Brain Area with Changed ReHo Values in Older vs. Younger Adults

ReHo values were analyzed by Alphasim correction (*p* < 0.05, cluster size > 146 voxels). Compared with younger adults, ReHo values of older adults decreased in the right superior temporal gyrus (STG), as shown in Figure 5.

#### 3.3.2. Correlation of ReHo Values and the Likelihood of Illusions

ReHo values of the STG were extracted according to the template in older adults. Correlation analysis between ReHo values and the likelihood of illusions in the STG in older adults showed that there was no statistical correlation between ReHo values and the likelihood of illusions in older adults (*p* > 0.05), as shown in Table 1 and Figure 5.

## 4. Discussion

The present study was conducted in younger and older adults with the same level of visual discrimination and explored the behavioral discrepancy between the two groups of participants in the condition of illusion. In addition, we also used ReHo values to compare the resting-state data of the two groups of participants, to find out the brain area of abnormal activity of older adults compared with younger adults, and to conduct correlation analysis with the likelihood of illusions of older adults. The behavioral results showed that the mean hit rates and *d’* of both younger and older adults in the F1 and F2 had no significant differences. However, the mean hit rates of both younger and older adults in F1B2 were significantly lower than those in F1B1, and F2B1 were significantly lower than those in F2B2, which manifested that all participants experienced the fission illusion and fusion illusion. Interestingly, both the mean hit rates and the *d’* showed that the fission illusion was significantly higher in older adults than in younger adults, but not in the fusion illusion. The results of neural activity manifested that the likelihood of the fission illusion had a positive correlation with ReHo values in the left MFG, and the likelihood of the fusion illusion had a positive correlation with ReHo values in the right IFG and right SFG. The results of the comparison between groups showed ReHo values of older adults decreased in the right STG, moreover, it was independent of the likelihood of illusions.

At the behavioral level, we found that all participants experienced obviously stable illusions of fission and fusion, which was consistent with previous studies [40,48,49]. What’s more, older adults had greater fission illusion, but the present study did not find discrepancy in the fusion illusion between younger and older adults, which indicated that the multisensory illusion of older adults was better than that of younger adults to some extent. These findings coincided with those of previous studies, for example, older adults were more vulnerable to the SiFI [18,19], however, there was no difference in the fusion illusion between the two groups [17]. Previous studies have found the fission illusion was lower in adult groups than in child groups, but the fusion illusion was semblable in both groups [50]. The fission illusion and fusion illusion may have different neural mechanisms, and become more manifested with ageing and development [6,25].

Laurienti et al., (2006) found that multisensory processing was enhanced in older adults to compensate for the decrease in unsensory ability and proposed the term of compensation mechanism [27]. Previous studies have manifested that the reason for older adults have greater multisensory illusion to rely on compensatory mechanisms [21]. In other words, when the information processing capacity of a single modality was insufficient, the strong SiFI could compensate for this defect by integrating the information of multisensory modalities [27]. For older adults, when the information on the visual modalities was insufficient to meet the needs of the judgment task, the information on the auditory modalities could be used as compensation to provide more environmental information [51]. However, this theory may not explain the results of the current study. In the current study, the level of visual discrimination was the same in older and younger adults, so older adults did not need greater multisensory illusion to compensate for visual discrimination.

Our results tended to support another explanation that for the enhancement of the SiFI in older adults was the decline of selective attention in older adults [52,53]. That’s a completely different explanation than the compensation mechanism, older adults were more likely to be distracted by distractions and thus make the wrong response [17]. Although Mishra and Gazzaley (2013) [54] found no age-related attention deficits in older adults, this may be related to their specific tasks, in their study, they used semantically congruent audiovisual stimuli rather than simple audiovisual stimuli, which could lead to more complex processing, in addition, the paradigm of multisensory redundancy effect was adopted, and there was no competition between visual and auditory modalities for attention resources. Poliakoff et al., (2006) [53] used a cross-modal conflict task to explore selective attention and ageing, and found that older adults were slower and made more mistakes on the tactile task with visual interference than younger adults. This suggests that older adults may be more easily distracted than younger adults, this suggested crossmodal attention deficit in older adults. According to perceptual load theory [55,56], in the case of low perceptual load, participants only need to consume part of their attention resources. Previous studies have found that selective attention ability of older adults decreased, so they could not ignore the interference of irrelevant stimuli, which led to part of the attention resources being occupied by irrelevant stimuli [53], this might explain the enhancement of the SiFI in older adults in the present study.

In older adults, we found that the spontaneous activity of several frontal regions (MFG, IFG and SFG) was significantly positively correlated with the SiFI. Previous studies have shown that fission illusion can show increased activity in the right frontal parietal attention network and STG [57,58] and left MFG [4,59], while the top-down influence of the frontal lobe region reflects the individual tendency to fusion illusion [10,15]. And compared with the no illusion trials, the illusion trials produced greater activation of angular gyrus [6,10,60], these results suggested that the frontal regions was an important activation region for the SiFI. In addition, the positive correlation between the likelihood of the SiFI and the activity of the frontal regions of older adults may be a result of cognitive aging. Previous studies have found that the left and right frontal lobes are engaged in executive function attentional control and inhibit automatic responses [61,62,63,64], and there was executive function aging in older adults [65]. The scaffolding theory of aging cognition posits that older adults add new neural pathways to compensate for impaired function and inefficient brain structures [66]. In our study, when the participants were in the F1B2 and F2B1 conditions, the auditory and visual signals were inconsistent, and older adults may have had to use more cognitive resources to pay attention to audiovisual stimuli, which may have led to an enhancement in frontal regions, but due to selective attention ability of older adults decreased, resulting in an enhancement in the SiFI.

The results of the comparison between older and younger adults manifested that ReHo values of the STG in older adults were decreased to varying degrees. The STG was considered to be a typical multisensory region [4,6,67]. Mishra et al., (2007) [16] used ERP technology to find that individuals who experienced a fission illusion exhibited greater negative components in the superior temporal gyrus at 110 and 130 ms. Previous studies have also found that fission illusion is linked to visual processing regions in the occipital gyrus [6,43,57,68]. However, the present study found that activity in the STG decreased in older adults, while the SiFI was enhanced. This was likely to suggest that the SiFI enhancement in older adults was not solely related to multisensory processing areas, this may be related to complex mechanisms resulting from factors such as executive function and selective attention. The present study demonstrated that when there was no difference in unisensory ability, the enhancement of multisensory illusion in older adults may not always be explained by compensation mechanisms.

## 5. Conclusions

In conclusion, our study manifested that when younger and older adults had semblable unisensory ability, the SiFI was stronger in older adults than in younger adults, and this did not seem to be explained by the compensation mechanism, ReHo values of rs-fMRI signals showed that the left MFG, right IFG and right SFG were significantly positively correlated with the SiFI in older adults. More importantly, the comparison between two groups showed that the weaken activity of the right STG of older adults was independent of the SiFI. Our findings demonstrated that when there was no discrepancy in unisensory ability, the enhancement of multisensory illusion in older adults may not always be explained by compensation mechanisms.

## Figures and Tables

**Figure 1 brainsci-12-01418-f001:**
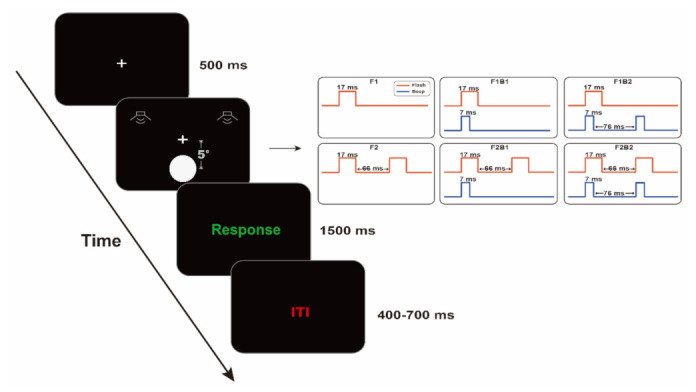
A schematic representation of experimental stimuli. There are six conditions, F1, F1B1, F1B2, F2, F2B1 and F2B2. F1 indicates the condition of presenting a flash, F1B1 indicates the condition in which a flash and a beep, F1B2 indicates the condition in which a flash and two beeps, F2 indicates that two flashes are present, F2B2 indicates that two flashes and two beeps at the same time, and F2B1 indicates the condition in which two flashes and a beep. The flash rendering time was 17 ms, and there was an interval of 66 ms between the two flashes. The beep rendering time was 7 ms, and there was an interval of 76 ms between the two beeps. At the beginning of each trial, the black screen showed a white “+” for 500 ms in the center, and the response screen showed 1500 ms, followed by 400–700 ms intertrial intervals (ITI). The red line indicates a flash, and the blue line indicates a beep.

**Figure 2 brainsci-12-01418-f002:**
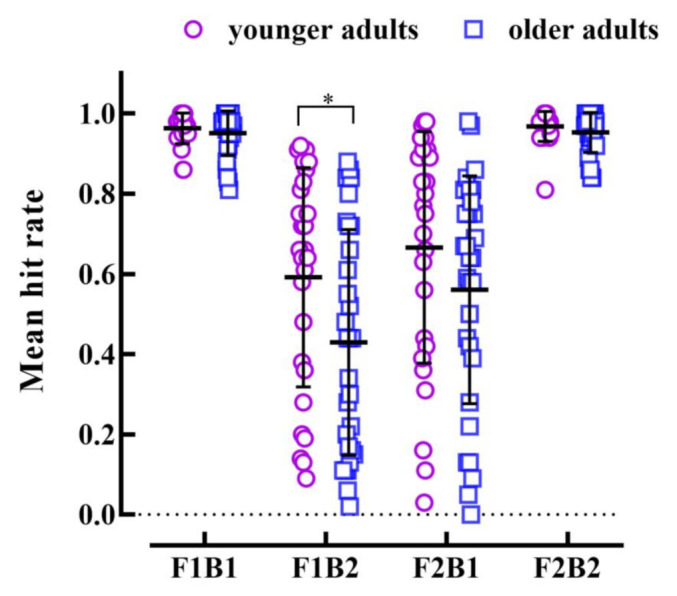
Mean hit rate of each participant in four conditions. F1B1 indicates the condition in which a flash is accompanied by a beep, F1B2 indicates the condition in which a flash is accompanied by two beeps, and the mean hit rate of F1B2 is significantly lower than that of F1B1, which produces the fission illusion. F2B2 indicates that two flashes are accompanied by two beeps at the same time; and F2B1 indicates the condition in which two flashes are accompanied by a beep. The mean hit rate of F2B1 is significantly lower than that of F2F2, which produces the fusion illusion. Short cross lines represent the mean hit rate, and error bars represent standard errors, * indicates *p* < 0.05.

**Figure 3 brainsci-12-01418-f003:**
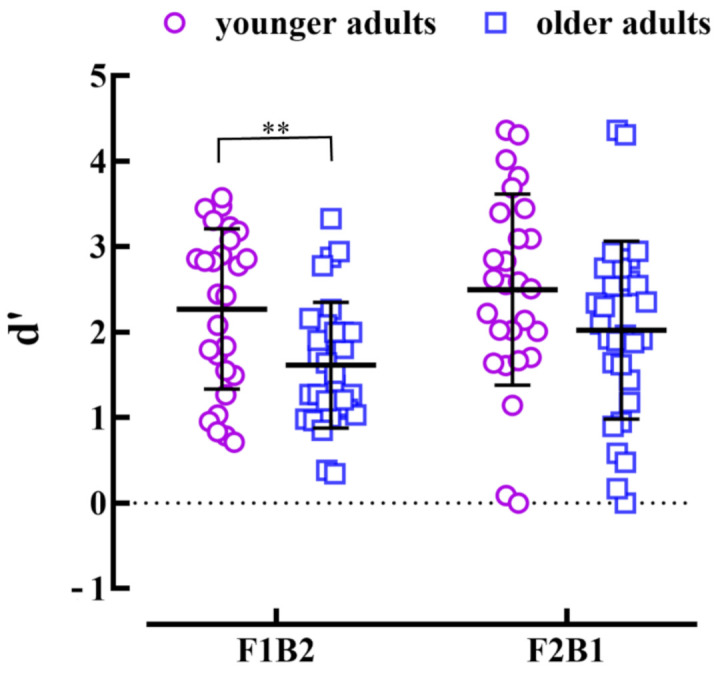
Mean *d’* of F1B2 and F2B1 conditions in younger and older adults. Short cross lines represent the mean hit rate, and error bars represent standard errors, ** indicates *p* < 0.01.

**Figure 4 brainsci-12-01418-f004:**
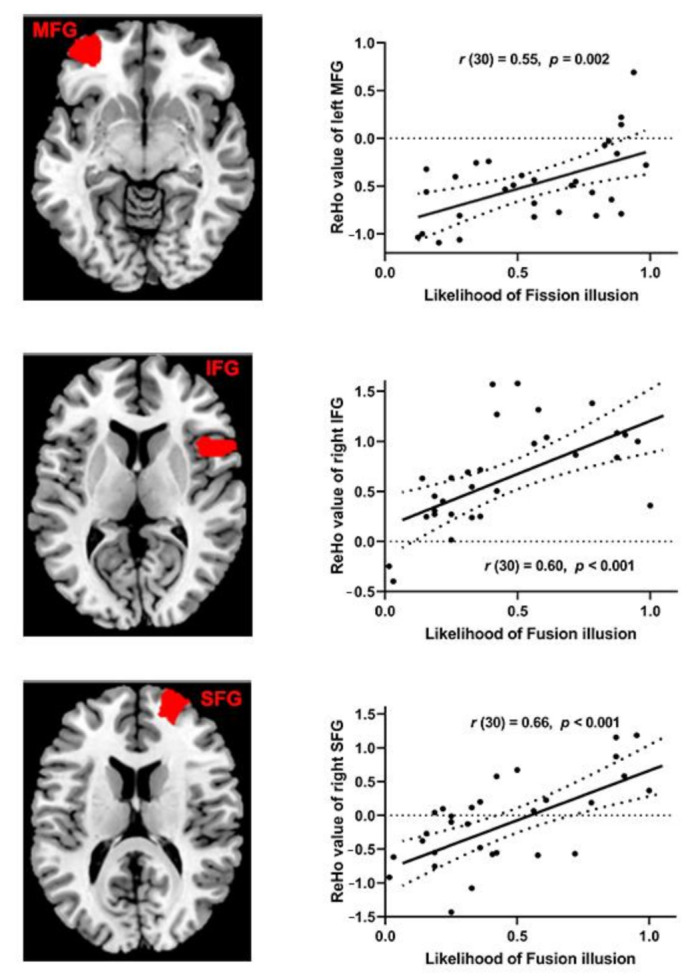
(**Left**). The older adults’ ReHo values and the likelihood of illusions showing a positive correlation in the left middle frontal gyrus (MFG), right inferior frontal gyrus (IFG) and right superior frontal gyrus (SFG). The red is the area where ReHo values are positively correlated with the likelihood of illusions. (**Right**). Scatter plot diagram shows the positive correlation between ReHo values of the left MFG and the likelihood of fission illusion of older adults, ReHo values of the right IFG and SFG and the likelihood of fusion illusion of older adults. Each point represents one participant’s data.

**Figure 5 brainsci-12-01418-f005:**
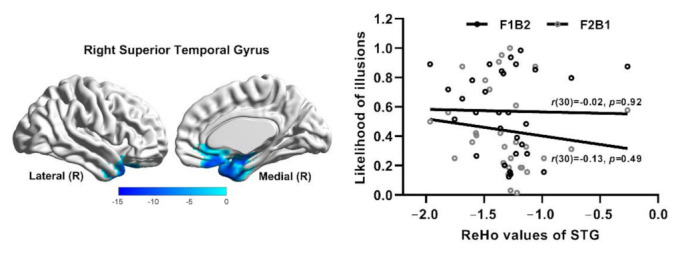
(**Left**). fMRI shows right superior temporal gyrus (STG) with changed ReHo values in older adults compared with younger adults, the blue is the area where ReHo values are significantly decreased. R = Right. (**Right**). Scatter plot diagram shows the correlation between ReHo values of the right STG and the likelihood of illusions of older adults. Each point denotes the data of one participant.

**Table 1 brainsci-12-01418-t001:** Correlation between ReHo values and the likelihood of illusions.

				MNI			Likelihood of Fission Illusion		Likelihood of Fusion Illusion	
	Brain Area	BA	Cluster Size	X	Y	Z	*r*	*p*	*r*	*p*
Older adults group	MFG	47	136	−33	51	−12	0.547	0.002 **		
	IFG	48	268	45	15	30			0.601	<0.001 ***
	SFG	6	397	18	3	54			0.657	<0.001 ***
Between-group comparisons	STG	36	1091	27	9	−36	−0.020	0.916	−0.132	0.488

Note: ** *p* < 0.01, *** *p* < 0.001.

## Data Availability

The data is available from the corresponding author upon reasonable request.

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
