# Peer review of "Compensation Mechanisms May Not Always Account for Enhanced Multisensory Illusion in Older Adults: Evidence from Sound-Induced Flash Illusion"

_brainsci, 2022, doi:10.3390/brainsci12101418_

Round 1

Reviewer 1 Report

The paper “Compensation mechanisms may not always account for enhanced multisensory illusion in older adults: evidence from Sound-induced flash illusion” investigates the role of age on the neural underpinnings of multisensory-based illusions. To this aim the authors analyzed resting state data from 30 older and 27 younger participants. The results showed that, with respect to younger participants, older participants had greater illusions and weaker brain activity in the superior temporal gyrus. The manuscript is timely and sound. The methodology is elegant and solid. The results are straightforward. The manuscript offers a general overview of the findings but it could be strengthened by further specifying the study's potential impact in a broader framework.

1) Methods – All participants were female. If this was the result of a specific inclusion criterium, and since previous studies about audio-visual illusions do not seem to consider gender as a main factor (Hess et al 2017 Cognitive Neuropsychiatry), it might be important to explicitly state it and to provide background for this choice.

2) Figure 5 – Adding the brain’s lateral view for the STG results might help the reader better localize the activation.

3) The manuscript correctly highlights the relationship between age, the temporal gyrus, and multisensory integration (audio-visual). To strengthen this claim, it could be beneficial to provide a mechanistic explanation of how/why the temporal gyrus neurally encodes multisensory integration. For example, it might be worth noting that the temporal gyrus plays a major role in audio-visual tasks, such as reading, including deficits associated with temporal hypoactivations in children, adolescents, young adults, and adults (Farah et al 2021, Frontiers in Psychology). On this basis, it could be proposed that the larger audio-visual bias and temporal hypoactivation in older than younger adults might be linked to age-related neural maturation of the temporal gyrus. What is the authors' account?

Reviewer 2 Report

The authors of this paper sought to explore the behavioral and neural evidence associated with age differences in the perception of the sound induced flash illusion when visual unisensory ability has been matched between age groups. The authors found that while both age groups experienced the fission and fusion portions of the illusion, the fission illusion was greater in older adults than in younger adults. Neuroimaging analysis revealed decreased regional homogeneity in the right superior temporal gyrus of older adults. They concluded that the enhanced multisensory illusory perception in older adults may not be always explained by compensation mechanisms when there is no difference in unisensory ability.

Overall, this study is very interesting and carefully designed. However, I feel that the introduction could be written to provide a stronger justification to the study. Furthermore, I am confused with the usage of the term “compensation mechanisms” in this paper. The authors described age-related decline in sensory processing effects on SIFI as compensatory, but in their discussion, while they explained that SIFI effects in the presence of matched unisensory ability may be due to age-related decline in selective attention, they don’t consider that as a compensatory mechanism. In other words, if the brain compensates for age-decline in sensory processing by an increase in multisensory integration, then an increase in multisensory integration caused by age-decline in selective attention can be termed as a compensation mechanism. I believe this confusion will be mitigated if the authors state what the instigator of the compensatory mechanisms is. For example, in the case of age-decline in unisensory processing, the mechanism of inverse effectiveness produces the observed perceptual effect. I will be glad if the authors can address this issue.

Below are specific comments:

Introduction

In line 58-60, the authors mentioned that studies have shown older adults can perceive the SIFI with longer stimulus onset… Considering that the aim of this study is centered on age effects on SIFI, the above sentence does not do justice to setting a proper background on the topic. A couple of relevant studies on this subject will help. I suggest that the authors talk about studies that investigated age effects on SIFI while explaining the gap and or additional information that their current study seeks to provide.

On line 66, authors write “The reason that older adults are better at multisensory illusion remains unknown.”. Authors should consider replacing the word “better” as it suggests that illusions are tests to assess perceptual performance. “More Susceptible” may be more appropriate

 It appears to me that it is not new information that the enhanced multisensory illusion effect in adults cannot be explained by only age-related decline in unisensory ability as presented by Hirst et al 2007 and 2019. Authors should consider strengthening the justification for this study.

Can authors elaborate on one study that used ReHo analysis to investigate sensory perception or cognition and if possible, in multisensory perception?

Method

Can the authors explain briefly the nature of the pre-experiment. Was it a visual detection task or temporal discrimination task? Although references were provided, it will be helpful to have a brief explanation when reading the paper to avoid having to go and read whole papers to understand what was done. Moreover, is there a justification for choosing 80% as a criterion for inclusion?

Studies have shown that reported hearing or manipulating auditory intensity of the stimulus can affect SIFI susceptibility (Hirsch 2019, Anderson 2004). If that is true, then compensation due to unisensory changes in adults may be explained by changes in auditory sensory processing. Is there a reason why authors did not match ability on auditory performance? And can authors discuss how this may or may not influence their interpretation?

How was normal vision or corrected normal vision determined?

How was normal hearing determined or were participants required to self-report their hearing ability?

Results

For the hit rate results for both fission and fusion, was there a reason why authors did not present the mean effect of age-group?

Figure legend. Can the authors define what the asterisk “*” represent?

For the signal detection theory analysis for the fusion condition in Figure 3, the two data points close to zero for the younger adults and the highest points for the older adults appear to be a bit distant from the others. While I am not a proponent of outlier detection, in some cases, it may be useful. Can the authors explain if they tried any method to determine if these points were outliers and what the results would be if yes.

Discussion

In line 414-417, the authors stated that “In the present study, as the selective attention ability of older adults decreased,”. This statement is a bit misleading, as it is suggesting that selective attention was manipulated in the study which doesn’t seem to be true. Can authors reword this sentence?

The authors explained their findings based on an age-related decline in selective attention. Since they refer to increased multisensory illusory perception arising from age-related sensory decline as a compensatory mechanism, it is confusing and inconsistent that similar perceptual effects arising from decline in selective attention not considered as a compensatory mechanism. Can authors clarify? (See general comment)

Round 2

Reviewer 1 Report

Accept

Reviewer 2 Report

I am very impressed with how the authors carefully answered each question with references to back their stands. Overall, I think this paper has been improved with the changes the authors have done and I recommend its acceptance for publication.